# Synergistic action of mucoactive drugs and phages against *Pseudomonas aeruginosa* and *Klebsiella pneumoniae*

Bingrui Sui,[1] Xiaoyu Li,[1] Na Li,[2] Yang Tao,[1] Lili Wang,[1] Yongping Xu,[1] Yumin Hou,[3] Bijie Hu,[2] Demeng Tan[3]

**ABSTRACT** *N*-acetylcysteine (NAC) and ambroxol hydrochloride (AMB) are commonly prescribed alongside antibiotics to alleviate sputum retention in lower respiratory tract infections, which are often caused by bacterial pathogens. With the rising threat of antibiotic resistance, phage therapy has emerged as a promising alternative alongside. However, no studies have explored the potential interactions between phages and these mucoactive agents despite their frequent concurrent use during phage therapy. Therefore, investigating the potential synergy and its subsequent impact on phage infection dynamics could enhance clinical strategies for treating bacterial infections with phages. Our study utilized *Pseudomonas aeruginosa* strain ZS-PA-35 and *Klebsiella pneumoniae* strain Kp36, alongside their respective phages, to investigate their interactions in the presence of NAC or AMB. Our findings indicate that, under specific conditions, these mucoactive agents can function as adjuvants to lytic phages, enhancing bacterial susceptibility to phage infection and facilitating subsequent phage proliferation. Our study revealed that these synergistic interactions are strongly influenced by the physiological characteristics of the phages, the surrounding microenvironments, and the physiology of host tissues, as varying outcomes of phage-host interactions were observed among different phages and across distinct media. Taken together, our results emphasize the complexity of interactions between phages and NAC or AMB, underscoring the need for caution when using combination treatments.

**IMPORTANCE** *N*-acetylcysteine (NAC) and ambroxol hydrochloride (AMB) are used in medical treatment of patients with acute and chronic bronchitis. Often, the choice of NAC or AMB is empirically determined by physicians. However, the potential impact of combining NAC or AMB with phage therapy remains unclear. To address this gap, a comprehensive understanding of their interplay is crucial to determine any potential synergistic effects. This study aims to elucidate how NAC or AMB influence phages targeting different receptors, thereby affecting their antibacterial activity against *Pseudomonas aeruginosa* and *Klebsiella pneumoniae*. Our results suggest that, under certain conditions, NAC or AMB provides an adjuvant effect by rendering the cells more susceptible to phage infection. These results contribute to advancing our understanding of the clinical combination of mucoactive agents and phage therapy, offering insights for optimizing treatment efficacy.

**KEYWORDS** *N*-acetylcysteine, ambroxol hydrochloride, bacteriophage therapy, phage-host interactions, antibiotic resistance

The global public health faces a critical challenge with the rapid escalation of multidrug-resistant (MDR) bacterial infections, which threatens the effectiveness of antibiotic treatments (1). Recently, the World Health Organization (WHO) highlighted this urgency by identifying a group of MDR-ESKAPE pathogens (*Enterococcus faecium*, *Staphylococcus aureus*, *Klebsiella pneumoniae*, *Acinetobacter baumannii*, *Pseudomonas*

Address correspondence to Xiaoyu Li, xiaoyuli@dlut.edu.cn, Bijie Hu, hu.bijie@zs-hospital.sh.cn, or Demeng Tan, demengtan@gmail.com.

The authors declare no conflict of interest.

See the funding table on p. 17.

*aeruginosa*, and *Enterobacter* species) that are the leading causes of nosocomial infections. This underscores the pressing necessity for research into novel antimicrobial therapies to combat MDR bacterial infections (2, 3). Phage therapy, the utilization of bacterial viruses known as bacteriophages (phages) for therapeutic purposes, has garnered renewed interest as a viable alternative to antibiotics, with high specificity in targeting and eliminating host bacteria effectively (4). Clinical successes of phage therapy have been observed across varying degrees in the treatment of bacterial infections, evidenced by numerous cases in Eastern Europe spanning decades, as well as recent compassionate care instances (5–9). Nevertheless, significant hurdles remain in the widespread acceptance of phage therapy, particularly concerning phage-host interactions in the presence of specific drugs. The potential synergistic or antagonistic effects of these interactions under such conditions still need to be thoroughly investigated (10).

Combining phages with antibiotics has demonstrated enhanced therapeutic effectiveness and is increasingly seen as a promising and practical approach to enhance the eradication of MDR bacteria, such as *Citrobacter amalonaticus*, *P. aeruginosa*, and *A. baumannii* (11–13). The synergy between phages and antibiotics, known as phage-antibiotic synergy (PAS), yields benefits far exceeding the individual effects of each component (14), which offers numerous advantages, including heightened bacterial-killing efficacy, decreased development of bacterial resistance, and enhanced ability to eradicate biofilms both in laboratory settings and within living organisms (14, 15). It is worth noting that beyond synergy, various interactions between phages and antibiotics have been observed, ranging from additive effects to antagonism or no discernible effect, depending on the specific antibiotics or phages involved (10). For instance, studies also raise concerns that such combinations can be variable and dependent on the antibacterial mechanism of action and stoichiometry (11). A recent study indicated that aminoglycosides produced by *Streptomyces* can impede an early stage of the phage life cycle, occurring before genome replication (16). Collectively, the combination of antibiotics with phages can result in a variety of outcomes, including synergistic or antagonistic effects. This highlights the need for more rigorous studies before considering *in vivo* administration.

Aside from antibiotics, *N*-acetylcysteine (NAC) and ambroxol hydrochloride (AMB) are commonly utilized mucoactive agents for treating lower respiratory diseases characterized by excessive airway mucus production, including cystic fibrosis and chronic obstructive pulmonary disease (17, 18). Moreover, NAC and AMB have acquired attention for their antimicrobial effects, in addition to their antioxidant and anti-inflammatory properties (19, 20). Also, studies have revealed that the combination of NAC with antibiotics can heighten bacterial susceptibility to antimicrobial agents (21–23). The precise mechanisms by which NAC attenuates pathogenic activity remain undetermined. Nonetheless, the bactericidal action of NAC might entail competitive inhibition of cysteine utilization or the interaction between its sulfhydryl groups and bacterial cell proteins (24). Moreover, several mechanisms have been proposed to elucidate how NAC disrupts extracellular polysaccharides (EPS), which are essential for biofilm formation (25). These mechanisms include the disruption of disulfide bonds within bacterial enzymes responsible for EPS synthesis and indirect interference with cellular metabolism. This interference subsequently reduces EPS production due to NAC's antioxidant properties (25, 26). On the other hand, AMB directly influences mucus composition by reducing its viscosity and promoting its expulsion, while exhibiting antimicrobial effects against *Candida albicans*, *C. parapsilosis*, and *P. aeruginosa* (27). Notably, AMB combined with the antifungal drug Fluconazole (FLC) has demonstrated synergistic effects against resistant *C. albicans* and biofilms at various developmental stages (28). The proposed synergistic mechanism involves AMB inhibiting the activity of drug transport pumps in resistant *C. albicans*, thereby elevating the intracellular concentration of FLC (28).

*P. aeruginosa* and *K. pneumoniae* have been identified as the predominant causative agents linked to lower respiratory ailments (29). At Shanghai Public Health Clinical

Center, informal reports and unpublished studies suggest that phage therapy using aerosolized phage cocktails in combination with NAC or AMB has been utilized for treating lung diseases. Despite this practice, no *in vitro* studies have ever investigated the interaction between phages and drugs targeting airway mucus hypersecretion. Questions remain regarding (i) whether concurrently administered NAC or AMB can impede the efficiency of phage therapy by eradicating their hosts and obstructing their progeny proliferation; and (ii) whether NAC or AMB-mediated bacterial colonization and virulence during critical stages of infection could modulate susceptibility to a range of phages, which necessitate further exploration . In this study, we systematically investigated their synergies in two different media and sought to determine whether synergistic or antagonistic effects prevail during these interactions. Our findings reveal that careful consideration should be given to the concurrent administration of phages with either NAC or AMB in clinical settings: (i) the physicochemical conditions of microenvironments may influence the efficiency and viability of phages in the presence of NAC or AMB, as we observed NAC-mediated phage degradation in LB medium; and synergistic variation between LB medium and cell culture medium; (ii) the effectiveness of combining NAC or AMB with phages hinges significantly on specific phage characteristics, such as receptor compatibility; and (iii) unexpected stochastic outcomes, such as NAC-mediated biofilm formation increasing and certain phages being unable to proliferate in cell culture medium, may arise in addition to the observed synergistic interactions. In light of these findings, careful consideration is essential in clinical decision-making. Further studies are needed to enhance the ability of clinicians to predict treatment outcomes, ultimately advancing the efficacy and optimization of therapeutic strategies for clinical applications.

## RESULTS

### Dose-dependent antimicrobial effects of NAC and AMB

First, we evaluated bacterial growth by exposing strains ZS-PA-35 and Kp36 to varying concentrations of NAC and AMB in LB medium, with NAC concentrations ranging from 20 µg mL$^{-1}$ to 2 mg mL$^{-1}$ and AMB concentrations from 25 µg mL$^{-1}$ to 2.5 mg mL$^{-1}$, respectively. Notably, the highest concentrations tested, NAC at 2 mg mL$^{-1}$ and AMB at 2.5 mg mL$^{-1}$, exhibited inhibitory effects on the growth of ZS-PA-35 and Kp36 throughout the entire experimental periods (Fig. S1, $P < 0.01$), indicating strong antibacterial properties of these mucoactive agents. It is noteworthy that NAC exhibited relatively stronger inhibition against *P. aeruginosa* compared to *K. pneumoniae*, while AMB was more effective in controlling the growth of *K. pneumoniae* than *P. aeruginosa*. We speculate that this intrinsic resistance, along with other antibiotic resistance mechanisms, may be critical as it can lead to various clinical outcomes. As expected, lower concentrations of NAC and AMB had negligible impact on bacterial growth, similar to the growth rates of the control strains. These results collectively show varying susceptibility profiles of *P. aeruginosa* and *K. pneumoniae* strains to elevated concentrations of NAC and AMB, and further corroborate earlier findings indicating that high concentrations of NAC and AMB possess antimicrobial effects, such as *Enterococcus faecalis* (30). For the assay combining phages and mucoactive agents detailed below, we selected concentrations of NAC at 200 µg mL$^{-1}$ and AMB at 250 µg mL$^{-1}$ to avoid inhibiting normal bacterial growth. This allowed for a precise investigation of the synergistic lytic effects between phages and NAC or AMB.

### Synergistic antimicrobial effects of phages combined with NAC or AMB in LB medium

Three diverse phages, phipa2, phipa10, and 117, targeting *P. aeruginosa* and *K. pneumoniae* by binding to key bacterial cell-surface receptors, including type IV pili (T4P), O-antigen, and colanic acid, respectively (Table S1), were strategically selected to evaluate their synergistic effects with NAC or AMB in LB medium, showcasing their potential for diverse applications (31, 32). As shown in Fig. 1A, the synergistic

antibacterial effect of combining NAC with phage phipa10 against strain ZS-PA-35 was significant (Fig. 1A, $P < 0.01$), with a noticeable enhancement after 1 h and a gradual increase over time until the end of the experiment. Likewise, a similar trend was observed in phage phipa10 titers when NAC was added, resulting in a more than twofold increase compared to phage phipa10 alone (Fig. 1E, $P$ value range from 0.0003 to 0.0211). Nevertheless, NAC had no significant impact on bacterial growth and phage production when combined with phages phipa2 or 117. The addition of NAC resulted in slightly increased phage susceptibility in strain ZS-PA-35 only at 2 h; however, regrowth occurred immediately later, rendering the NAC-treated culture even resistant to phage phipa2. In contrast, phage 117 titers decreased compared to the control without NAC, while bacterial growth remained unaffected (Fig. 1B, C, F and G).

In clinical trials, phage cocktails are rationally formulated to widen their host range based on their receptors (33–35). A cocktail comprising phages phipa2 and phipa10, often recognized for their efficacy against various bacterial isolates other than mono-phage, is commonly utilized to minimize the regrowth of resistant populations during phage therapy. Thus, we investigated whether NAC could enhance the efficacy of the phage cocktail (phages phipa2 + phipa10). Because strain ZS-PA-35 is susceptible to both phage phipa2 and phage phipa10, we utilized two phage receptor mutants for plaque assay: a *pilT* mutant, resistant to phage phipa2 but susceptible to phage phipa10, and a mutant containing the *galU* deletion, allowing quantification of phage phipa2 while being resistant to phage phipa10. By employing these two mutants, we precisely assessed the dynamics of phage composition in the cocktail when exposed to strain ZS-PA-35 in the presence or absence of NAC. As anticipated, we observed a significant synergistic effect between NAC and the phage cocktail, primarily attributed to the positive interaction between phage phipa10 and NAC. Specifically, the titer of phage phipa10 increased 2.8-fold in the NAC-treated cultures compared to the control without NAC, whereas the titer of phage phipa2 remained unaffected by the addition of NAC (Fig. 1D, $P < 0.05$; and Fig. 1H, $P < 0.01$).

Unlike NAC, the combined application of AMB and phage phipa2 demonstrated a synergistic inhibitory effect on ZS-PA-35, but not with phage phipa10 (Fig. 2A and E), phage 117 (Fig. 2C and G), or the phage cocktail (phages phipa2 + phipa10) (Fig. 2D and H) with their corresponding hosts. For phage phipa2, this combination significantly inhibited bacterial growth after 2 h of incubation, resulting in consistently lower bacterial densities compared to phage phipa2 alone. A similar trend was observed in the titers, with a twofold increase compared to phage phipa2 alone (Fig. 2B and F, $P < 0.01$). Overall, NAC and phage phipa10 exhibited synergistic effects either alone or in the phage cocktail format, while AMB showed synergy with phage phipa2 when used alone in LB medium. Taken together, this suggests that NAC or AMB could serve as alternative

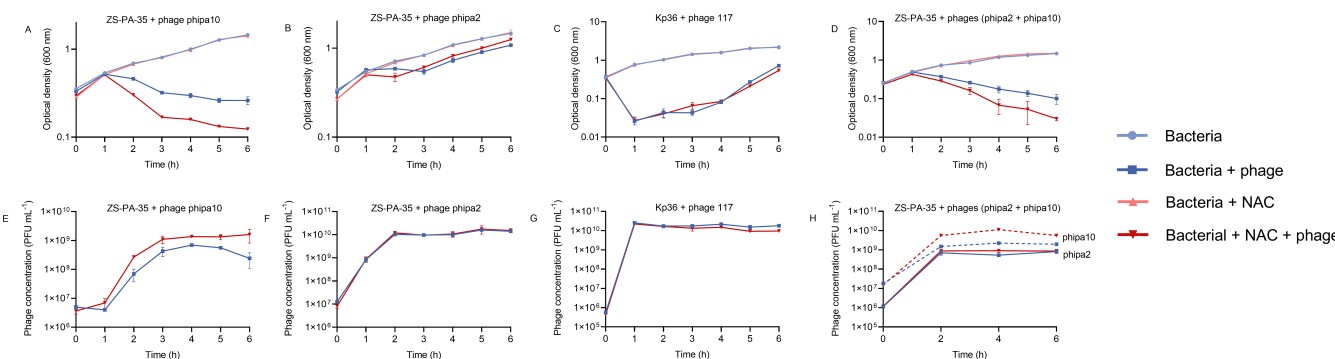

**FIG 1** The effect of NAC on phage-host interaction dynamics in LB medium over 6 h. Panels (A–D) show the inhibition curves of phage-host interactions with and without NAC. The conditions include phage phipa10 with ZS-PA-35 (MOI = 0.001), phage phipa2 with ZS-PA-35 (MOI = 0.001), phage 117 with Kp36 (MOI = 0.0001), and a combination of phages phipa2 and phipa10 with ZS-PA-35 (MOI = 0.1). The results are compared to controls without phages. Panels (E–H) display phage production under the same conditions for phages phipa10, phipa2, 117, and the combination of phages phipa2 and phipa10 with ZS-PA-35. Data represent mean ± SD from triplicate experiments.

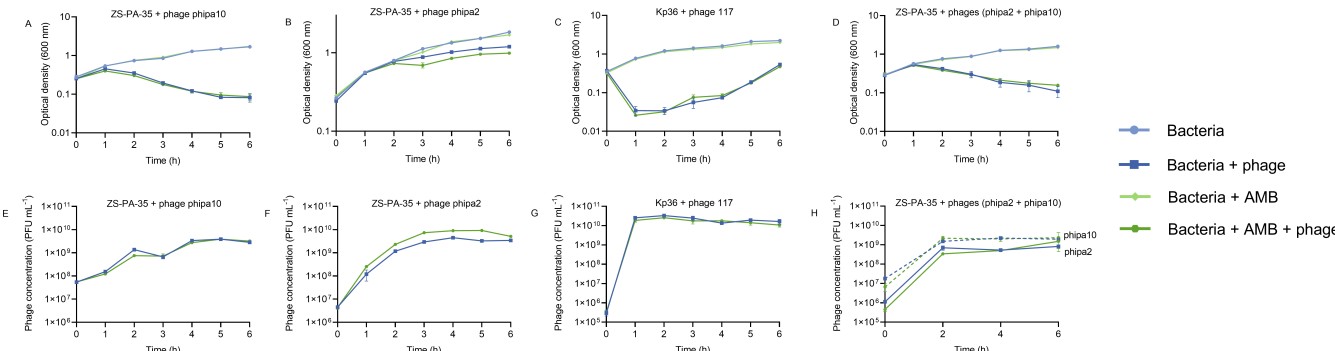

**FIG 2** Impact of AMB on phage-host interaction dynamics in LB medium over 6 h. Panels (A–D) show the inhibition curves of phage-host interactions with and without AMB under the following conditions: phage phipa10 with ZS-PA-35 (MOI = 0.01), phage phipa2 with ZS-PA-35 (MOI = 0.001), phage 117 with Kp36 (MOI = 0.0001), and a combination of phages phipa2 and phipa10 with ZS-PA-35 (MOI = 0.1). Panels (E–H) display the corresponding phage production under these conditions. Data are presented as mean ± SD from triplicate experiments.

adjuvants for phage treatment of MDR bacterial infections, while also demonstrating a high degree of specificity toward their phage-host interactions.

## Effect of NAC and AMB on phage-host interactions in cell culture medium

While we observed the synergistic effects of NAC or AMB with phage phipa10 or phage phipa2 in LB medium, discrepancies may arise due to the inherent differences in complex microenvironments during phage administration. Our understanding of the interactions between phages and their hosts in human environments remains incomplete. To better mimic the *in vivo* conditions following phage entry into the bloodstream, we conducted the above experiments using the cell culture medium. This approach is expected to unveil their biological aspects and elucidate their relevance in a human context. Specifically, strain ZS-PA-35 demonstrated a slower growth rate in cell culture medium compared to that observed in LB medium. The $OD_{600}$ increased from 0.2 to approximately 1.3 over a 24-h period. Following the addition of phage phipa10, $OD_{600}$ decreased to ~0.177 within 6 h. However, the emergence of a subpopulation subsequently grew exponentially, reaching an average $OD_{600}$ of ~1.16 at 24 h. A similar growth trend was observed in phage phipa10 cultures combined with or without NAC or AMB, despite the addition of NAC resulted in a slightly lower cell density compared to phage phipa10 alone at 24 h (Fig. 3A). An intriguing finding is that the proliferation of phage phipa10 was markedly inhibited in the cell culture medium, irrespective of whether NAC or AMB was present. Within 24 h, the phage concentration dropped by more than four orders of magnitude from $4.64 \times 10^9$ to $1.43 \times 10^5$ PFU mL$^{-1}$ (Fig. 3D). Similarly, upon the addition of phage phipa2 to ZS-PA-35, the growth curve of bacteria closely mirrored that of the ZS-PA-35 strain alone. The $OD_{600}$ of the NAC-added cultures reached 1.06 after 24 h, slightly lower than the control (Fig. 3B). The release of phage phipa2 with NAC peaked at approximately $9 \times 10^8$ PFU mL$^{-1}$ by 18 h, marking a 1.64-fold increase compared to the wild-type (WT) strains. Similarly, the addition of AMB resulted in slightly increased phage production, reaching levels of $1.2 \times 10^9$ PFU mL$^{-1}$ at 12 h and $9.4 \times 10^8$ PFU mL$^{-1}$ at 18 h, which was approximately 1.88 and 1.7 times higher than that of phage phipa2 alone, respectively (Fig. 3E). However, no significant effect was observed in terms of bacterial growth during these periods (Fig. 3B).

For *K. pneumoniae*, strain Kp36 exhibited exponential growth in cell culture medium with an increase in $OD_{600}$ from 0.3 to ~1.9 in the first 6 h and finally stabilized at ~2.8. Notably, with the addition of phage 117, the bacterial density was significantly dropped to 0.14 within 6 h. This was followed by the exponential regrowth of a subpopulation likely resistant to phage 117 infection, reaching an $OD_{600}$ of 2.55 after 24 h. Unlike in LB medium, the addition of NAC enhanced bacterial susceptibility to phage 117 at 12 h, resulting in an $OD_{600}$ of 0.91, which was lower than that observed with phage 117 alone

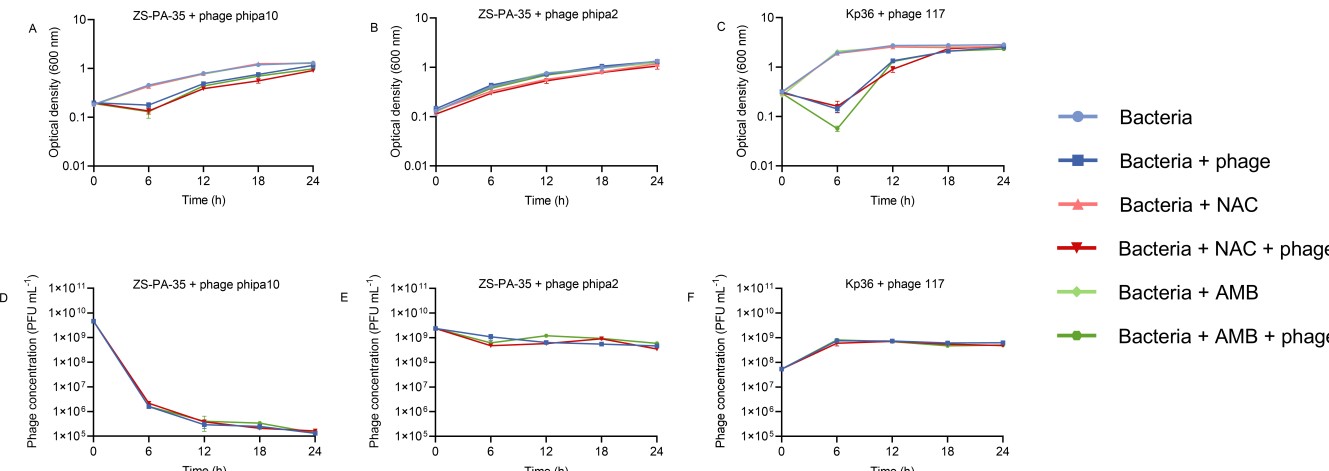

**FIG 3** Effect of NAC and AMB on phage-host interaction dynamics in cell culture medium over 24 h. Panels (A–C) illustrate the inhibition curves of phage-host interactions with and without the addition of mucoactive drugs (NAC and AMB) under the following conditions: phage phipa10 with ZS-PA-35 (MOI = 1), phage phipa2 with ZS-PA-35 (MOI = 1), and phage 117 with Kp36 (MOI = 0.01). Panels (D–F) present the corresponding phage production under these conditions. Data are shown as mean ± SD from triplicate experiments.

(Fig. 3C, $P < 0.01$). In the presence of AMB, the bacterial density reached 0.05 at 6 h, showing a slight reduction compared to phage 117 alone (Fig. 3C). However, regrowth occurred, eventually reaching a level comparable to that of phage 117 alone. Despite both NAC and AMB demonstrating a synergistic effect with phage 117 in cell culture medium, there was no change in the amount of phage production, which may appear paradoxical (Fig. 3F). Conceptually, this may be attributed to NAC- or AMB-mediated increased susceptibility to accelerate phage lysis, eventually achieving similar phage production.

## Exposure to NAC decreases phage stability

The net phage concentration in the media is determined by both phage production and any phage loss due to degradation caused by the addition of NAC or AMB. To improve our understanding of phage pharmacokinetics, we examined the impact of NAC or AMB on phage stability. In doing so, phage concentrations in the test media were continuously measured for a 6-h period, with their stability confirmed using plaque assay (Fig. 4). In LB medium, the NAC treatment resulted in a significant reduction in the phage stability within 6 h, with the stability of these phages (phipa10, phipa2, and 117) decreasing by approximately 96.55%, 99.66%, and 96.38%, respectively (Fig. 4A through C). On the contrary, the addition of AMB had no effect on phage stability, except that the stability of phage 117 was slightly reduced after 6 h (Fig. 4D through F). Furthermore, phages remained stable in cell culture medium with 2% fetal bovine serum (FBS) (Fig. 5A through C), except that phage phipa2 decreased by 28% and 21% at 2 and 4 h, respectively, and phage 117 decreased by 15% from the starting point, with the addition of AMB (Fig. 5E and F, $P < 0.05$). Yet, the omission of FBS caused varying alterations in the stability of phages in the presence of NAC or AMB. Interestingly, the addition of NAC increased the stability of phage phipa2 more than 200%, extending it from 2 to 6 h (Fig. 5H, $P$ value range of 0.0013–0.0286). Likewise, the addition of AMB gradually increased the efficacy of phage 117 to almost 200% at 6 h (Fig. 5L, $P$ value range of 0.0019–0.0499). The stability of phage phipa10 exhibited a decline of approximately 45% or 36% at 6 h compared to its initial levels in cultures treated with NAC or AMB, respectively (Fig. 5G and J, $P < 0.05$). Collectively, these findings suggest that phages exhibit instability in LB medium supplemented with NAC, while they remain relatively stable in cell culture medium supplemented with 2% FBS. These observations align with other known factors, such as pH fluctuations, which are particularly harmful to phages. Low pH environments,

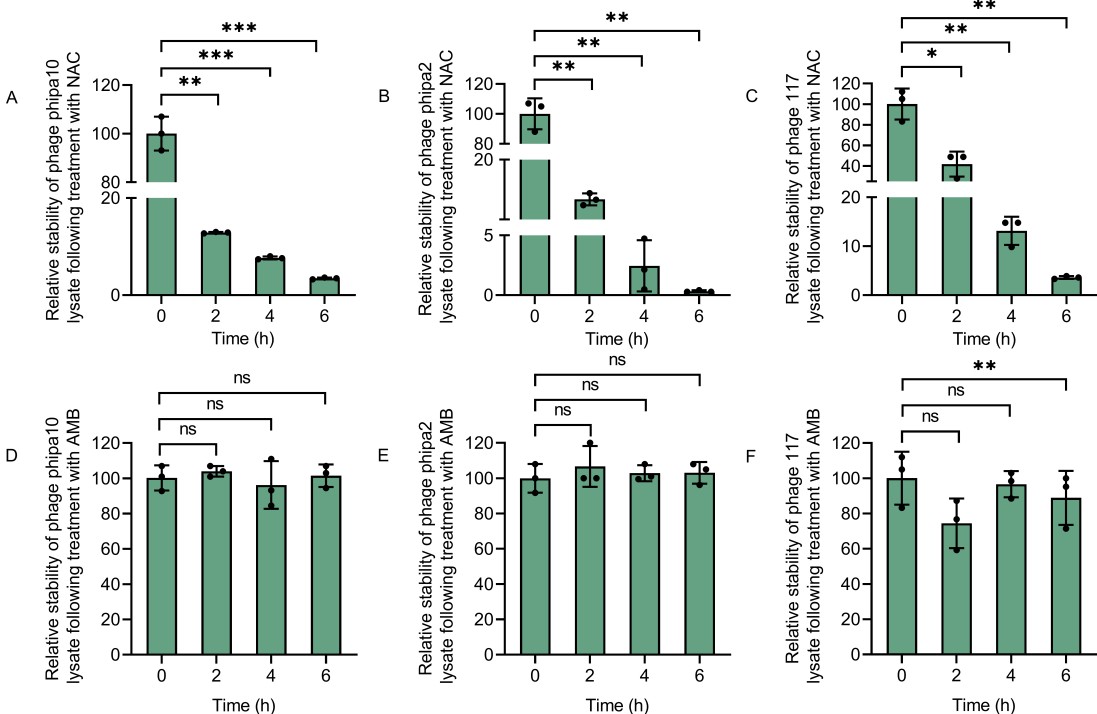

**FIG 4** Effects of NAC and AMB on the stability of phages phipa10, phipa2, and 117 in LB medium over 6 h. Panels (A–C) display the impact of NAC on the stability of phages phipa10, phipa2, and 117, respectively. Panels (D–F) demonstrate that phages phipa10, phipa2, and 117 maintain stability in LB medium supplemented with AMB. Data are presented as mean ± SD from triplicate experiments. Statistical significance is indicated as *$P < 0.05$, **$P < 0.01$, and ***$P < 0.001$.

like those found in the stomach, are especially detrimental to phages (36). Therefore, the route of phage administration, especially with NAC, is crucial for lung infections.

## Exposure to AMB affects the stability of eukaryotic virus

Recent advancements in sequencing technology have unveiled that phages are the predominant components of the gastrointestinal virome, collectively known as the phageome (37). They are also considered to have a significant impact on the composition and functionality of the human gut microbiome in both health and disease (38). Our findings may indicate that NAC or AMB has the capacity to influence the phage composition. However, whether NAC or AMB possess antiviral effects against other eukaryotic viruses remains unknown. To address this, we used baby hamster kidney (BHK)-21 cells to detect plaque formation of human coronavirus OC43 (OC43) with and without NAC or AMB treatment. As shown in Fig. S2, the enumeration of the plaque formation of OC43 indicated that the addition of NAC had no significant antiviral effect. This result is consistent with the study by Garozzo et al., where NAC alone did not exhibit any antiviral activity, and the survival rate increased to 100% only when combined with Oseltamivir (39). Conversely, AMB demonstrated a time-dependent effect on plaque formation within 6 h of incubation, resulting in a decrease in plaque count from $4.78 \times 10^5$ PFU mL$^{-1}$ to $1.77 \times 10^3$ PFU mL$^{-1}$ within 6 h, which was approximately 30 times lower than the $6 \times 10^4$ PFU mL$^{-1}$ after 6 h of incubation alone (Fig. S2B). These results suggest that AMB may possess antiviral properties. Similarly, Yang et al. demonstrated that AMB could significantly inhibit the proliferation of influenza A/Aichi/68 (H3N2) virus and improve the survival rate of mice in a mouse experiment (40). Thus, the route of administration, whether through aerosolization, intravenous injection, or oral ingestion, may influence the viral composition, including phages and eukaryotic viruses,

## Cell-culture medium with 2% FBS

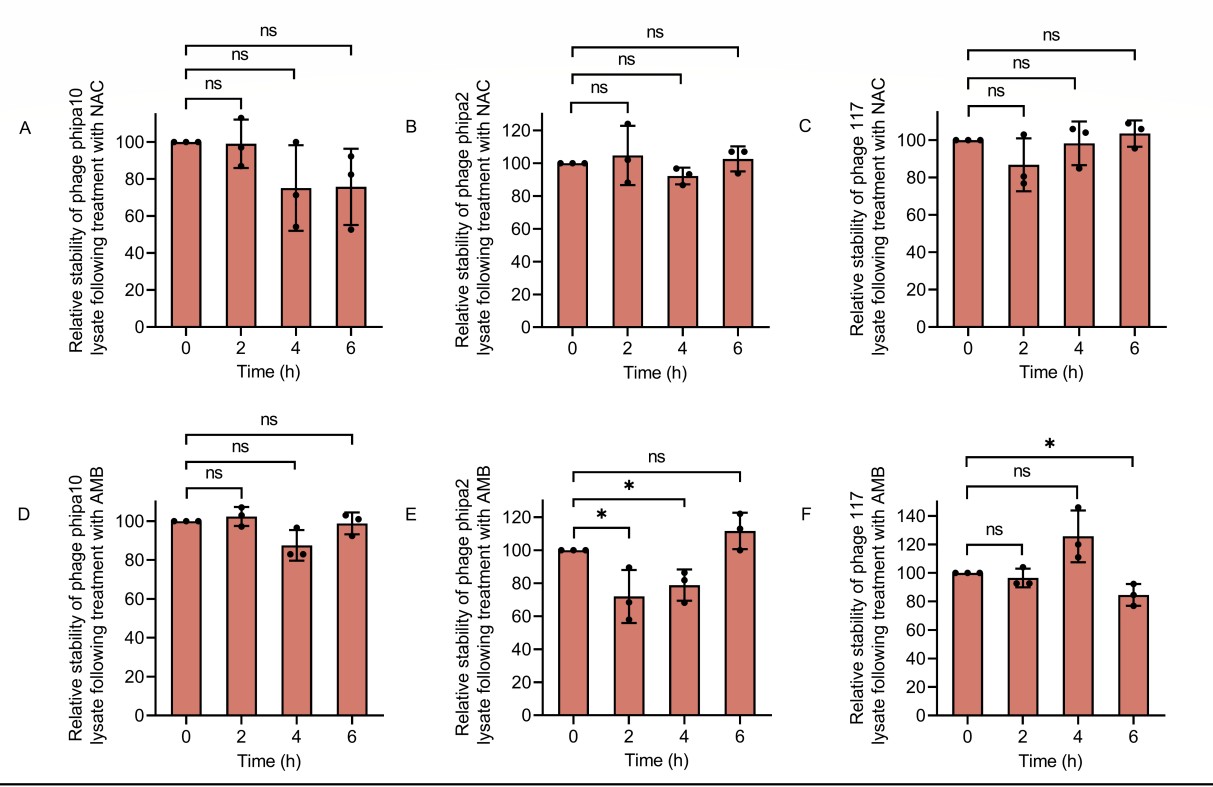

## Cell-culture medium without 2% FBS

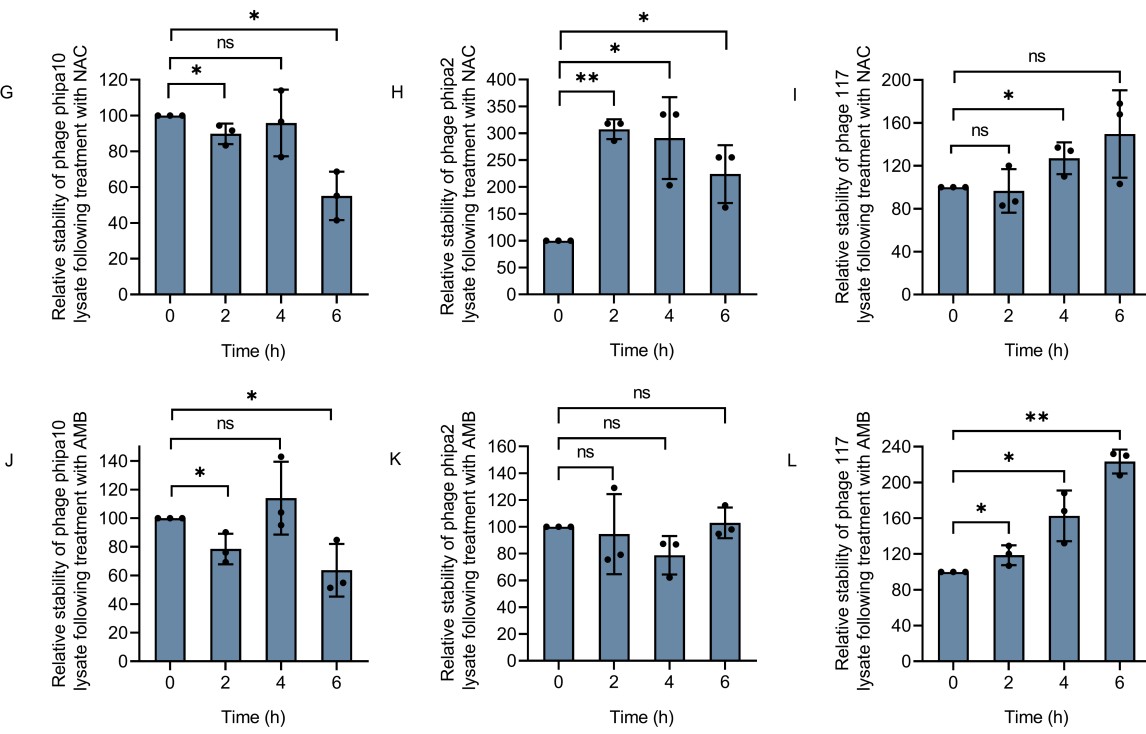

**FIG 5** Stability of phages phipa10, phipa2, and 117 in cell culture medium with or without 2% FBS in the presence of NAC or AMB over 24 h. Panels (A–F) represent the stability of phages in cell culture medium supplemented with 2% FBS. Panels (A–C) show the effects of NAC on the stability of phages phipa10, phipa2, and 117, respectively, while (D–F) display the effects of AMB on the same phages. Panels (G–L) depict the stability of phages in cell culture medium without 2% FBS. Panels (G–I) illustrate the effects of NAC on phages phipa10, phipa2, and 117, respectively, and panels (J–L) demonstrate the effects of AMB on the same phages. Data are presented as mean ± SD from triplicate experiments. Statistical significance is indicated as *$P < 0.05$ and **$P < 0.01$.

as observed in this study. This variability arises because the antiviral effectiveness of NAC or AMB can fluctuate depending on the composition of the medium.

## Effect of NAC/AMB on phage adsorption and phage receptor expression

The increase in phage susceptibilities and production primarily hinges on the phage infection process, particularly the likelihood of phage-host interaction (41). Common mechanisms of bacterial resistance to phages include altering, losing, or masking phage receptors (42). NAC or AMB predominantly impacts bacterial surface structure, potentially influencing phage receptors (43, 44). To investigate the effect of NAC or AMB on phage adsorption receptors and their expression levels, we conducted phage adsorption and quantified gene expression assays (Fig. 6 and 7). The adsorption rates of phage phipa10 and phage phipa2 were slightly increased by the addition of NAC or AMB to strain ZS-PA-35, respectively, though the changes were not statistically significant. Whereas notable impacts of both NAC and AMB were observed on strain Kp36 for phage 117 (Fig. 6). In the presence of NAC, the adsorption rate of phage 117 increased by 1.54-fold at approximately 9 min (Fig. 6C). On the other hand, AMB increased the adsorption rate of phage 117 by 2.3-fold at 3 min of adsorption (Fig. 6F). Next, we investigated whether the alterations in phage adsorption were because of NAC or AMB on the expression of phage receptors on the bacterial surface or the removal of physical barriers between the phage receptor and the phage binding protein. As both NAC and AMB could result in an increased adsorption rate of phage 117 to strain Kp36. However, gene quantification assays revealed that in *P. aeruginosa* ZS-PA-35, AMB enhances the expression of *galU* (Fig. 7D) and stimulates *pilT* expression mainly at high cell density, specifically at an $OD_{600}$ of 1.0, indicating a possible density-dependent effect (Fig. 7E). Nevertheless, at both bacterial densities, with $OD_{600}$ of 0.3 and 1.0, neither NAC nor AMB exerted any influence on the expression of the gene *wcaJ* in *K. pneumoniae* Kp36 (Fig. 7C and F). Based on our findings, there appears to be some inconsistency between the quantification of the phage adsorption process and gene expression. One possible

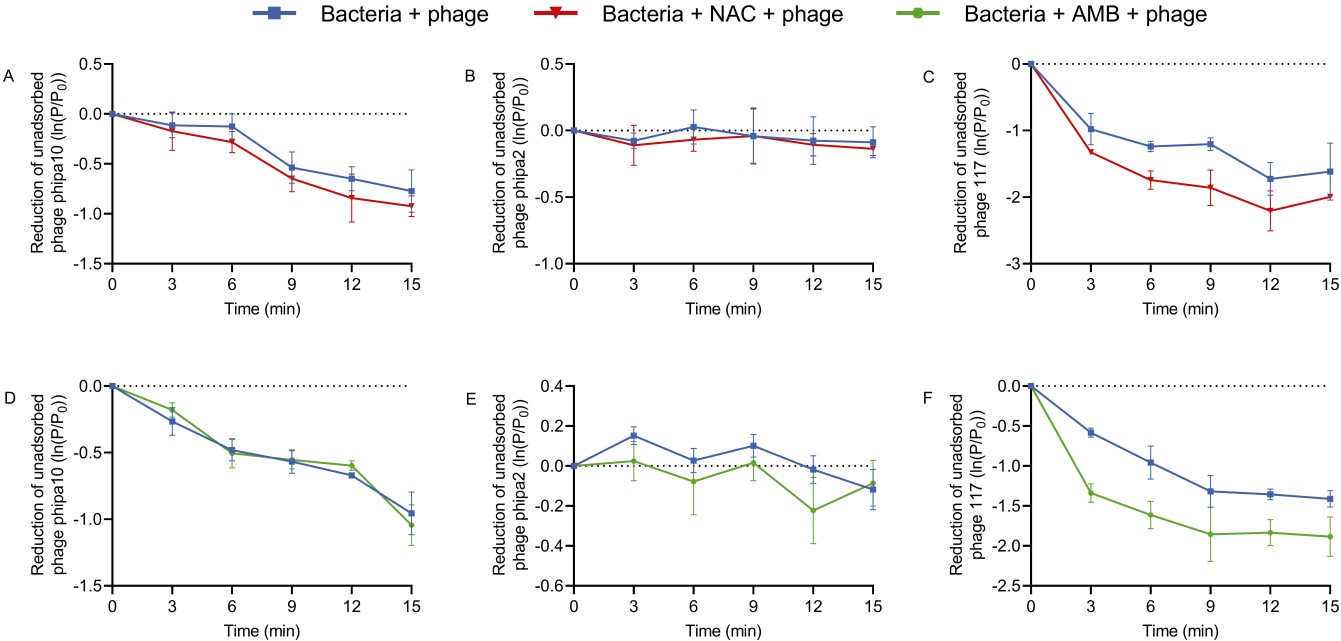

**FIG 6** Effects of NAC and AMB on the adsorption of phages phipa10, phipa2, and 117 to their respective hosts. Phage adsorption assays were performed to evaluate the influence of NAC and AMB on the adsorption efficiency of phages at an MOI of 0.001. Free phage titers were quantified after brief incubation with each phage-host combination, in the presence or absence of NAC or AMB. Panels (A–C) show the effects of NAC on the adsorption of phage phipa10 to ZS-PA-35, phage phipa2 to ZS-PA-35, and phage 117 to Kp36, respectively. Panels (D–F) illustrate the effects of AMB on the adsorption of phage phipa10 to ZS-PA-35, phage phipa2 to ZS-PA-35, and phage 117 to Kp36, respectively. Error bars indicate standard deviations from triplicate experiments.

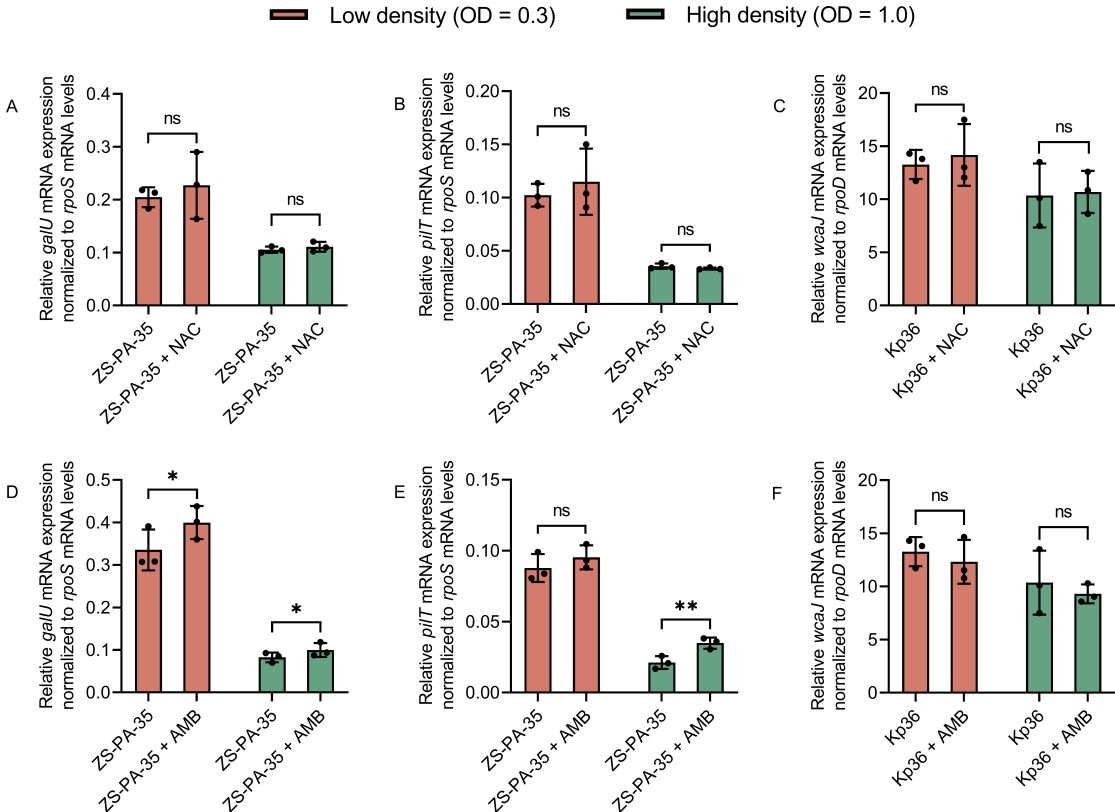

**FIG 7** Phage receptor gene expression quantification assay. Panels (A and B) illustrate the expression of *galU* and *pilT* genes relative to the reference gene *rpoS* in WT- and NAC-treated ZS-PA-35 at OD 0.3 and OD 1.0, respectively. Panels (D and E) depict the expression under AMB treatment. Panels (C and F) show the expression of the *wcaJ* gene relative to *rpoD* in WT and Kp36 with added NAC or AMB at OD 0.3 and OD 1.0. Error bars represent mean ± SD among triplicate samples, with *, indicating statistical significance at $P < 0.05$ and ** at $P < 0.01$.

explanation is that the observed effects on bacterial surface receptors are specific to certain receptors and vary depending on the experimental conditions. This specificity may be due to AMB subtly altering the bacterial surface structure rather than enhancing receptor expression. Thus, the exposure of phage adsorption receptors on the bacterial surface facilitates phage attachment to host bacteria, thereby increasing the potential for successful phage lytic cycles as bacterial density rises.

## NAC promotes bacterial biofilm formation

Phage receptors, including O-antigen, colanic acid, and T4P, play essential roles in biofilm formation and bacterial motility. Biofilms are complex structures composed of surface-attached cells encased in a matrix of extracellular polysaccharides (EPS), extracellular DNA (eDNA), and cellular debris (45). The formation of biofilms by *P. aeruginosa* and *K. pneumoniae* is a critical factor in their pathogenicity, as these biofilms contribute to their ability to colonize biomaterials and confer resistance to antimicrobial agents (46). Notably, NAC has been shown to disrupt biofilm growth and interfere with multiple stages of biofilm formation (47). In this context, we assessed the biofilm-forming capabilities of *P. aeruginosa* strain ZS-PA-35 and *K. pneumoniae* strain Kp36 following extended exposure to NAC or AMB. Surprisingly, we found that the application of NAC resulted in significant promotion of biofilm formation in both strains, with a 1.84-fold and 2.4-fold increase, respectively, in comparison to the control (Fig. 8, *P* values of 0.0008 and 0.03, respectively). In contrast, Guerini et al. reported that a dosage of 0.5 mg mL⁻¹ of NAC reduced the formation of *P. aeruginosa* biofilm by approximately 64% (48). Conversely, AMB exhibited no influence on biofilm formation in either strain (Fig.

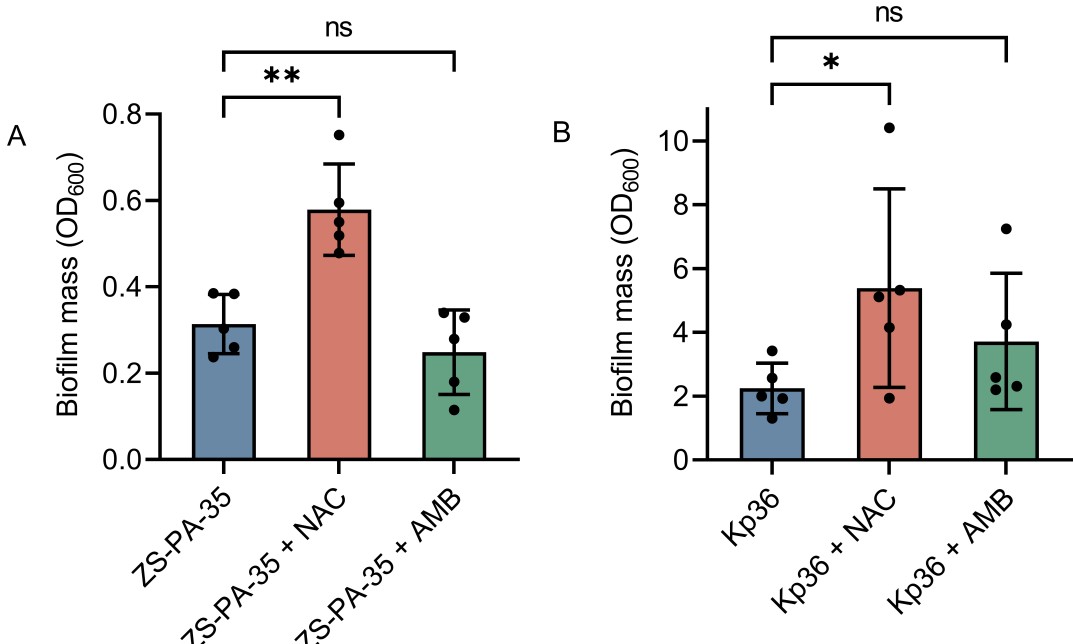

**FIG 8** Biofilm formation assay of strains ZS-PA-35 (A) and Kp36 (B). Biofilm formation was assessed with and without of NAC or AMB, and the absorbance of biofilm biomass was quantified using 0.4% crystal violet staining. Error bars indicate standard deviations, with experiments conducted in triplicate. Statistical significance is denoted by *, where * means $P < 0.05$ and ** means $P < 0.01$.

8), which is inconsistent with the results reported by Cataldi et al. that AMB has an anti-biofilm function and that AMB inhibits all stages of biofilm formation (49). For the phage phipa2 receptor, T4P plays an essential role in facilitating twitching motility (50). We proceeded to assess the impact of NAC or AMB on bacterial twitching abilities. The twitching motility assay indicated that the strain's movement distance on LB agar plates containing NAC or AMB resembled that of the strains alone, suggesting that NAC or AMB did not significantly affect T4P-mediated twitching motility (Fig. S3). Overall, our findings contradict previous research, as both NAC and AMB have been shown to inhibit biofilm formation (24, 27). This underscores the significant intraspecific variations observed in biofilm formation and twitching motility in both *P. aeruginosa* and *K. pneumoniae*. We hypothesized that this difference could be due to the lower concentrations of NAC or AMB used, suggesting that these lower concentrations might not be suitable for clinical use in preventing bacterial colonization.

## DISCUSSION

Pharmacological strategies for managing airway hypersecretion involve several categories of mucoactive agents, including expectorants, mucoregulators, and mucokinetics (51). Traditional mucolytic medications like NAC function by reducing the viscoelasticity of mucus through the reduction of disulfide bonds, aiding in its clearance (52). In addition, expectorants like AMB modify mucus consistency by promoting surfactant production and regulating the secretion of serous and mucus fluids, thereby enhancing the effectiveness of coughing in the respiratory tract (53). Furthermore, both NAC and AMB demonstrate anti-inflammatory and antioxidant properties, along with antimicrobial activity against various bacterial strains, including opportunistic respiratory pathogens like *K. pneumoniae* and *P. aeruginosa* (30, 54).

Extensive *in vitro* and *in vivo* studies have explored the mechanisms underlying the beneficial effects of NAC and AMB, revealing their potential to enhance treatment outcomes (18, 53, 55, 56). These investigations have shown promising results, particularly in patients with respiratory conditions like community-acquired pneumonia, when

these mucoactive agents are used in combination with antibiotics or antiviral therapies (52, 57). However, the excessive use of antibiotics has contributed to the emergence of *P. aeruginosa* and *K. pneumoniae* as leading nosocomial pathogens, responsible for infections associated with significant rates of mortality and morbidity (58). These infections present a considerable challenge for treatment due to the reduced responsiveness to antimicrobial agents and the recurrent emergence of antibiotic resistance (59). In response to these challenges, phage therapy is being evaluated as a feasible option for both therapeutic and prophylactic purposes, offering an alternative to traditional antibiotic approaches (60). However, the practical implementation of phage therapy in clinical settings is hindered by interactions with other drugs, such as antibiotics and mucoactive agents like NAC and AMB (10, 61). It is increasingly evident that specific antibiotics can have both positive and negative effects on the outcome of phage-host interactions, necessitating careful consideration of whether there is a synergistic relationship between phages and antibiotics (10). As a result, ongoing research is needed to focus on understanding and harnessing the synergistic effects between phages and NAC or AMB, as well as their regulation of phage-host interactions and other phenotypic traits, to explore the potential clinical benefits of phage therapy.

In this study, we present evidence for a synergistic effect between NAC and phage phipa10 in *P. aeruginosa* strain ZS-PA-35, as demonstrated by the increased susceptibility to the phage in LB medium. This finding offers novel insights into the potential of NAC as a phage enhancer. However, when assessing the impact of NAC on other phages, such as *P. aeruginosa* phage phipa2 and *K. pneumoniae* phage 117, the addition of NAC did not significantly affect phage-host interactions, including bacterial growth and phage proliferation, compared to the control medium. Interestingly, the addition of AMB appeared to render strain ZS-PA-35 more susceptible to phage phipa2, but did not alter the behavior of phages phipa10 or 117. One of the most striking observations from this study was the differential stability of the phages in the presence of NAC versus AMB. All phages tested exhibited reduced stability in the presence of NAC, as evidenced by a significant decrease in phage concentration in LB medium, whereas phages remained stable in cell culture medium. We hypothesize that this loss of stability is due to the sulfhydryl group in NAC, which may disrupt the disulfide bonds of phage proteins, leading to the decomposition of phages into smaller peptide fragments. In contrast, AMB did not induce similar destabilizing effects, suggesting that its mode of action differs from that of NAC. When evaluated in cell culture medium, the combined effects of NAC or AMB with the phages differed substantially from the results observed in LB medium. Notably, both NAC and AMB exhibited a synergistic effect with phage 117, and all phages showed relative stability in the cell culture environment. The variations in progeny release observed across different media highlight the complex interplay of factors influencing phage activity, suggesting that medium composition and environmental conditions play a pivotal role in determining the outcome of phage-host interactions. These findings underscore the importance of considering a range of factors, such as the properties of the phages, the characteristics of the microenvironment, and the presence of mucoactive agents, when developing phage-based therapies. The apparent synergism between phages and mucoactive agents like NAC and AMB may be dependent on these factors, and a deeper understanding of these interactions is essential for translating laboratory-based findings into effective clinical applications. Thus, to effectively deliver active phages for treating infections, particularly in the context of lung infections, it is crucial to account for these variables in the design of future therapeutic strategies..

## Potential mechanisms underlying synergistic effect

Although both NAC and AMB have been widely used in clinical practice for several decades as common treatments for bronchopneumonia, their precise antimicrobial mechanisms remain poorly understood (55). The sulfhydryl component of NAC functions as an antioxidant to neutralize free radicals, thus aiding the immune system in breaking down both inter- and intra-molecular disulfide bonds in bacterial proteins (62). Unlike

NAC, AMB has anti-inflammatory properties, which it achieves by suppressing the activity of neutrophils at various stages. It also enhances mucociliary clearance by affecting the function of ion channels and transporters in the airway epithelium. These include Na/HCO₃ and Na/K/Cl cotransporters, the cystic fibrosis transmembrane conductance regulator, and aquaporins (49). The apparent contradiction between NAC and AMB regarding phage susceptibility and phage production may relate to their antimicrobial mechanisms discussed above. We hypothesize that the enhanced susceptibility of phages toward strain ZS-PA-35 may result from NAC or AMB facilitating the circumvention of the physical barrier between the phage-binding protein and the phage receptor. By removing these barriers, such as EPS and alginate, would facilitate the phage access to the host cells. In support of our hypothesis, Li et al. showed that AMB inhibited the alginate production of *P. aeruginosa* in a dose-dependent manner (63). However, the fact that AMB does not enhance the breakdown of alginate, implies its potential role in reducing the production of this exopolysaccharide. This finding is further supported by a study showing that AMB markedly decreased both the protein expression and activity of guanosine diphospho-dD-mannose dehydrogenase. This suggests that the drug may modulate the transcription of the algU operon gene, although the precise molecular mechanism remains to be elucidated (63). Additionally, studies have suggested that quorum sensing (QS) and AMB may work simultaneously to stimulate the production of T4P (54). Indeed, phage receptors, T4P and O-antigen, respond to QS signals positively, and manipulation of QS can therefore affect phage sensitivity (64). This feature of the QS and mucoactive agents has made it difficult to understand how specific regulation of phage receptor can be extracted from each side, how the QS and mucoactive agents might drive distinct output behaviors, and, in turn, how NAC and AMB attenuate QS-regulated phenotypes in bacterial communities. Although QS is often viewed as a beneficial process that allows bacteria to adapt to new hostile environments, a likely explanation for this apparent contradiction between QS-regulated T4P and AMB's antagonistic QS properties is that NAC bound to LasR and RhlR proteins in a similar manner to the autoinducer cognate, thus attenuating QS-upregulated phage receptor phenotype. Further investigation is needed to understand how bacteria integrate QS and AMB to regulate T4P production dominantly.

## Medium composition affects viral stabilities

Despite clear evidence that both NAC and AMB exhibit synergistic efficacy in eradicating microbial pathogens when combined with specific phages in LB medium, little is known about the impact of microenvironmental factors on their antimicrobial and antiviral efficacy, particularly their interactions within human cell culture medium. It is crucial to evaluate their performance to ensure reliable results. This will enable us to translate our findings directly into the clinical application of phage therapies. Since the LB medium may not be an appropriate methodology for evaluating their efficacy intended for use in clinical environments, studying phage-host interactions synergistic effect with NAC or AMB within a cell culture environment is likely to resemble the natural cellular environment found in the body, mimicking the metabolic profile of human plasma. We believe the knowledge gained from experiments like this would suit the therapeutic purpose. Undoubtedly, the impact of external factors affecting antimicrobial materials remains poorly understood.

To shed light on this crucial issue, we examined how variations in medium composition affect the stabilities of tested phages and OC43 Influenza viruses. One notable finding from our assays was that during a 6-h incubation period, all phages demonstrated minimal decay when incubated in the cell culture medium compared to LB medium. This suggests that the degrading impact of NAC may be counteracted by the components present in the cell culture medium. However, it is important to note that our assessment only covers a short timeframe, and longer-term decay in the cell culture medium cannot be ruled out. Moreover, both phage phipa10 and phage phipa2 were unable to proliferate in strain ZS-PA-35 in the cell culture medium, challenging

the previous concept of phage therapy that phages exploit bacterial cells for survival and proliferation. Remarkably, our findings underscore the potential of AMB in treating OC43. Prior research has indicated that AMB shares lysosomotropic properties, meaning it can penetrate and accumulate within late endosomes and lysosomes, potentially impeding viral replication within cells (65). These results emphasize the necessity of considering the infectious microenvironment, given the vast and diverse microbiome within the human body, which can influence the interactions between phages, NAC, AMB, and pathogens. Consequently, we propose that human cell culture medium can alter stability and susceptibility to pathogens, challenging assumptions based solely on standard laboratory conditions.

## NAC affects biofilm formation

For *P. aeruginosa* and *K. pneumoniae*, biofilm formation is critical for the proper initial colonization stage and is becoming more well-acknowledged as a passive virulence factor that facilitates several infectious disease processes (66). Both O-antigen and colanic acid, known as extracellular polymer, serve as molecular catalysts to facilitate its exploration and inhabitation of diverse niches and contribute to biofilm formation (67). Within *P. aeruginosa* there are examples of both NAC-activated and NAC-repressed biofilm formation (68, 69). Several studies have shown that, disrupting preformed biofilms (both initial and mature), and decreasing the survival of bacteria within biofilms (24, 47, 70). Human serum with NAC raised the expression of some biofilm-associated genes and enhanced the biosynthesis of extracellular matrix by *S. aureus* ATCC25923 and *P. aeruginosa* PAO1 (71). However, we show that NAC increased biofilm formation in *P. aeruginosa* strain ZS-PA-35, as well as *K. pneumoniae* strain Kp36. Thus, large variations in NAC-mediated biofilm formation exist between different strains of *P. aeruginosa* and *K. pneumoniae*.

In conclusion, this study provides valuable insights into the interplay between mucoactive agents (NAC and AMB) and phage-host interactions, highlighting a synergistic effect between phages and NAC or AMB against MDR *P. aeruginosa* and *K. pneumoniae* under specific conditions. Our findings underscore the variability in the impact of mucoactive agents, even among isolates of the same bacterial species, such as *P. aeruginosa* and *K. pneumoniae*. Several limitations of the study warrant consideration: (i) the use of only two clinical isolates, which may not fully capture the phenotypic diversity of bacterial populations; (ii) the common co-administration of antibiotics with phage therapy, underscoring the importance of investigating potential synergistic effects among phages, mucoactive agents, and antibiotics; and (iii) the reliance on laboratory conditions, highlighting the necessity for clinical trials to validate these findings in real-world settings. Future research should focus on comparing the efficacy of sequential versus simultaneous therapeutic strategies to minimize antagonistic interactions, as well as elucidating the mechanisms underlying synergistic and antagonistic effects in combinatorial treatments. These efforts will enhance the clinical applicability of phage-based therapies in combination with mucoactive agents, addressing critical challenges posed by MDR pathogens.

## MATERIALS AND METHODS

### Bacterial strains and phages

Strains and phages used in this study are listed in Table S1. *P. aeruginosa* ZS-PA-35 and *K. pneumoniae* Kp36 were grown in LB Miller medium (10 g $L^{-1}$ peptone, 5 g $L^{-1}$ yeast extract, and 10 g $L^{-1}$ NaCl) or on 1.5% LB agar plates under aerobic conditions at 37°C. Three lytic phages used in this study were previously identified and characterized along with their hosts (31, 32). Phages were propagated using the double-layer plaque assay as previously described and stored at 4°C until further use (72). Both NAC and AMB (Sigma-Aldrich, St. Louis, MO, USA) were freshly prepared in sterile distilled water and filtered through a 0.22-µm-pore membrane.

## Cell line stocks

BHK-21 fibroblast cells (clone 13; ATCC CCL-10) and rhabdomyosarcoma (RD) cells (ATCC CCL-136) were cultured in high-glucose Dulbecco's modified Eagle medium (DMEM; Cytiva, Marlborough, MA, USA) supplemented with 10% FBS (Lonsera, Uruguay), 100 IU mL$^{-1}$ penicillin, and 100 µg mL$^{-1}$ streptomycin, kept at 37°C with 5% $CO_2$. The virus OC43 (ATCC VR-1558) was cultured in RD cells line for a period of 3–4 days at 37°C with 5% $CO_2$ in high-glucose DMEM, which was further 2% FBS-supplemented. Virions were obtained by centrifuging the culture media at 3,000 × $g$ for 30 min, to remove cell debris. The virus stocks were then stored at −80°C until use.

## Bacterial susceptibility to NAC/AMB in LB medium

Overnight cultures of ZS-PA-35 and Kp36 strains were diluted 1:1,000 in 15 mL of LB medium. NAC and AMB stock solutions were introduced to achieve final concentrations ranging from 20 µg mL$^{-1}$ to 2 mg mL$^{-1}$ for NAC and 25 µg mL$^{-1}$ to 2.5 mg mL$^{-1}$ for AMB. The cultures were incubated at 37°C with agitation, alongside a parallel control group without NAC or AMB. Bacterial growth was assessed by measuring optical density at $OD_{600}$, with hourly sampling over an 8-h period. All experiments were conducted in triplicate. The highest concentrations of NAC or AMB that did not adversely affect bacterial growth were identified and used in subsequent studies.

## Phage-NAC/AMB combination assay

Overnight cultures of ZS-PA-35 and Kp36 strains were diluted 1:1,000 in 15 mL of LB medium. NAC or AMB were added to achieve final concentrations of 200 and 250 µg mL$^{-1}$, respectively. The cultures were grown to an $OD_{600}$ of 0.3, corresponding to approximately $2.4 \times 10^8$ CFU mL$^{-1}$, before being infected with 10 µL aliquots of phage phipa2, phage phipa10, a phage cocktail (phipa2 + phipa10), or phage 117 at low MOIs < 0.1. Parallel control cultures without NAC or AMB served as experimental controls. Bacterial growth was monitored by measuring $OD_{600}$ at 1 h intervals over 6 h. Concurrently, phage titers in the LB medium were quantified by plaque assay to assess infectious phage concentrations. A 0-h baseline was established following a 10-min incubation with phages under shaking conditions. All experiments were performed in triplicate.

For phage-NAC/AMB killing assays in cell culture medium, time-killing assays were performed with some modifications from the previous protocol. Briefly, overnight cultures of strains ZS-PA-35 and Kp36 were diluted 1,000-fold in 10 mL of high-glucose DMEM with 2% FBS, supplemented with NAC and AMB to the previously mentioned final concentrations, and incubated at 37°C until reaching an $OD_{600}$ of 0.2. For strain ZS-PA-35, 100 µL aliquots of phage phipa10 (~$4.64 \times 10^{10}$ PFU mL$^{-1}$) and phage phipa2 (~$2.38 \times 10^{10}$ PFU mL$^{-1}$) was added at an MOI of 1. For strain Kp36, a 100 µL aliquot of phage 117 ($5.32 \times 10^8$ PFU mL$^{-1}$) was added at an MOI of 0.01. Parallel control cultures without treatment were used as controls. Bacterial growth and quantification of infectious phages were monitored every 6 h over a 24-h period. All experiments were conducted in triplicate.

## The impact of mucoactive agents on phage stability

To test the stability of phage phipa2, phage phipa10, and phage 117 in the presence of NAC or AMB, 5 µL phage suspensions was added to 1 mL of LB medium, high-glucose DMEM with 2% FBS, and high-glucose DMEM, each containing 200 µg mL$^{-1}$ NAC or 250 µg mL$^{-1}$ AMB. Control cultures without NAC or AMB were also included. The titer of phages was determined at 2-h intervals over a 6-h period using the double-layer agar assay.

## Antiviral activity of NAC/AMB on OC43 assay

The antiviral efficacy of NAC and AMB against respiratory coronavirus OC43 was assessed using plaque assays on BHK-21 fibroblast cells, following established protocols (73). Initially, BHK-21 fibroblast cells ($5 \times 10^5$ cells per well) were seeded into six-well plates (Corning, NY, USA) and incubated at 37°C with 5% $CO_2$ for 18 h to form a confluent monolayer. OC43 virus suspensions (50 µL) were combined with 450 µL of high-glucose DMEM, and NAC or AMB was added to achieve final concentrations of 200 or 250 µg $mL^{-1}$. After incubating at 37°C for various time points (0, 2, 4, and 6 h), the mixtures were serially diluted and applied to BHK-21 fibroblast cell monolayers, which were then incubated for an additional 4 h. Subsequently, the supernatant was replaced with DMEM containing 2.4% Avicel and 2% FBS, and the infected cells were further cultured for approximately 4 days until plaques became visible. The cells were then fixed with 10% formaldehyde and stained with 1% crystal violet to visualize plaques. Viral titers were quantified as plaque-forming units per milliliter (PFU $mL^{-1}$). A control group without NAC or AMB treatment was included for comparison.

## Phage adsorption assay

Overnight bacterial cultures were grown to an $OD_{600}$ of 0.3 at 37°C after being back-diluted 1,000 times in 15 mL of LB medium containing either NAC or AMB at final concentrations of 200 or 250 µg $mL^{-1}$. Phages were then added at an MOI of 0.001, and aliquots were collected at 3-min intervals over a 15-min period. After centrifugation at $16,600 \times g$ for 90 s at 4°C, the number of phage particles in the supernatant was determined using plaque assays, calculated as previously described by Tan et al. (32). Specifically, absorption rates were calculated as the slope of the equation: $y = (\ln(P_0) - \ln(P))/B_0$, where $P_0$ and $B_0$ represent the total number of plaques and bacteria at time 0, respectively, and $P$ represents the number of free plaques at each time point. The average of three separate experiments was used for these calculations.

## Real-time quantitative PCR

Overnight bacterial cultures were grown to $OD_{600}$ of 0.3 and 1.0 at 37°C after being back-diluted 1,000 times in LB medium containing NAC or AMB at final concentrations of 200 or 250 µg $mL^{-1}$. Total RNA was extracted using TRIzol reagent (Thermo Fisher Scientific) following the manufacturer's instructions (74). Residual genomic DNA was removed using RNase-Free DNase I (Thermo Fisher Scientific). The RevertAid First Strand cDNA Synthesis Kit (Thermo Fisher Scientific) was used to synthesize cDNA according to the manufacturer's protocol. During real-time PCR, target genes were amplified using primers listed in Table S2. The expression levels of *galU* and *pilT* in strain ZS-PA-35 relative to the reference gene *rpoS*, and *wcaJ* in strain Kp36 relative to the reference gene *rpoD* were determined using SsoAdvanced SYBR Green Supermix (Bio-Rad) as described previously (75). Each experiment was conducted three times, with each run performed in triplicate.

## Biofilm formation and twitching motility assay

Biofilm formation was examined in a 12 mL polystyrene tube through the method of crystal violet, as previously described (31). In short, 3 µL of bacterial cultured overnight was incorporated into 3 mL of LB medium with 200 µg $mL^{-1}$ NAC or 250 µg $mL^{-1}$ AMB, or control without either. The combination was kept statically at 37°C to promote biofilm development. After a week, every tube was carefully cleaned three times with flowing water before being dyed with 5 mL of 0.4% crystal violet. Following the 30 min of staining, each tube's excess stain was removed, and 5 mL of 75% ethanol was used to dissolve the crystal violet that had attached itself to the bacterial cells. Biofilm creation was evaluated by crystal violet under $OD_{600}$, with a control group comprised of just LB medium. Twitching motility was evaluated by subsurface stab tests on LB plates as earlier reported (31). In summary, a 2 µL aliquot of overnight bacterial culture with a final

concentration of NAC (200 µg mL$^{-1}$) or AMB (250 µg mL$^{-1}$) was punctured and inoculated at the center of the plate (1.5% agar), and with parallel controls without NAC or AMB. After 72 h of incubation at 37°C, the agar was discarded and the plates that remained were statically for 30 min at room temperature applying with 0.4% crystal violet, rinsing with plain water removed excessive stains. It was established where the twitching zone was between the agar and petri dish.

## Statistical analysis

All graphics and statistical analyses were performed using GraphPad Prism version 9.2 (GraphPad, La Jolla, CA). Results are presented as mean ± standard deviation (SD). Significance levels were evaluated using one-way analysis of variance for multiple-group comparisons and Student's $t$ test for two-group comparisons. A $P$ value of<0.05 was considered statistically significant.

## ACKNOWLEDGMENTS

This project was supported by the Fundamental Research Funds for the Central Universities (DUT23YG214) to X.L., the National Natural Science Foundation of China (NSFC) (No. 42376099) to D.T., the National Natural Science Foundation of China (No. 82072325) to B.H., the Shanghai Sailing Program (No. 23YF1443700), and the Clinical Research Plan of the Shanghai Municipal Health Commission (No. 20224Y0287) to N.L.

## AUTHOR AFFILIATIONS

[1]MOE Key Laboratory of Bio-Intelligent Manufacturing, School of Bioengineering, Dalian University of Technology, Dalian, China
[2]Department of Infectious Diseases, Zhongshan Hospital, Fudan University, Shanghai, China
[3]Shanghai Public Health Clinical Center, Fudan University, Shanghai, China

## AUTHOR ORCIDs

Xiaoyu Li ⓘ http://orcid.org/0000-0002-5618-8888
Yongping Xu ⓘ https://orcid.org/0000-0002-3946-9223
Bijie Hu ⓘ http://orcid.org/0000-0002-8687-992X
Demeng Tan ⓘ http://orcid.org/0000-0002-9204-833X

## FUNDING

| Funder | Grant(s) | Author(s) |
| --- | --- | --- |
| MOST \| National Natural Science Foundation of China (NSFC) | 82072325 | Bijie Hu |
| MOST \| National Natural Science Foundation of China (NSFC) | 42376099 | Demeng Tan |
| Shanghai Sailing Program | 23YF1443700 | Na Li |
| Shanghai Municiapl Health Commission | 20224Y0287 | Na Li |
| Fundamental Research Funds for the Central Universities | DUT23YG214 | Xiaoyu Li |

## AUTHOR CONTRIBUTIONS

Bingrui Sui, Conceptualization, Data curation, Formal analysis, Investigation, Methodology, Validation, Writing – original draft, Writing – review and editing | Xiaoyu Li, Funding acquisition, Methodology, Project administration, Supervision, Writing – original draft, Writing – review and editing | Na Li, Conceptualization, Investigation, Methodology, Resources, Visualization | Yang Tao, Investigation, Methodology | Lili Wang, Investigation, Methodology | Yongping Xu, Investigation, Methodology, Project administration,

Resources | Yumin Hou, Investigation, Methodology | Bijie Hu, Conceptualization, Data curation, Funding acquisition, Project administration, Resources, Supervision.

## DATA AVAILABILITY

The bacteria and phage genomes can be assessed in GenBank with the subsequent identification numbers: *P. aeruginosa* ZS-PA-35 (GCA_020567355.1), *K. pneumoniae* Kp36 (CP047192), phage phipa2 (OK539824.1), phage phipa10 (OK539826.1), and phage 117 (MN149903.1).

## ADDITIONAL FILES

The following material is available online.

### Supplemental Material

**File S1 (Spectrum01601-24-s0001.xlsx).** Data sources for statistical analyses.
**Supplemental Material (Spectrum01601-24-s0002.docx).** Figures S1 to S3; Tables S1 and S2.

### Open Peer Review

**PEER REVIEW HISTORY (review-history.pdf).** An accounting of the reviewer comments and feedback.

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
