## [Reviewer comments · Microbiology Spectrum]

Microbiology Spectrum

Synergistic Effect of Combination Mucoactive Drugs with Phages against *Pseudomonas aeruginosa* and *Klebsiella pneumoniae*

Bingrui Sui, Xiaoyu Li, Na Li, Yang Tao, Lili Wang, Yongping Xu, Yumin Hou, Bijie Hu, and Demeng Tan

Corresponding Author(s): Demeng Tan, Fudan University

Review Timeline:

Submission Date:

July 2, 2024

Accepted:

November 26, 2024

Editor: Victor Gonzalez

Reviewer(s): Disclosure of reviewer identity is with reference to reviewer comments included in decision letter(s). The following individuals involved in review of your submission have agreed to reveal their identity: Joseph Atia Ayariga (Reviewer #1)

Transaction Report:

DOI: <https://doi.org/10.1128/spectrum.01601-24>

Re: Spectrum01601-24 (Synergistic Effect of Combination Mucoactive Drugs with Phages against *Pseudomonas aeruginosa* and *Klebsiella pneumoniae*)

Dear Dr. Demeng Tan:

Your manuscript has been reviewed by two reviewers who agree that it merits its publication in *Microbiology Spectrum*. As suggested by the reviewers, minor corrections should be made before publication.

Your manuscript has been accepted, and I am forwarding it to the ASM production staff for publication. Your paper will first be checked to make sure all elements meet the technical requirements. ASM staff will contact you if anything needs to be revised before copyediting and production can begin. Otherwise, you will be notified when your proofs are ready to be viewed.

PubMed Central: ASM deposits all *Spectrum* articles in PubMed Central and international PubMed Central-like repositories immediately after publication. Thus, your article is automatically in compliance with the NIH access mandate. If your work was supported by a funding agency that has public access requirements like those of the NIH (e.g., the Wellcome Trust), you may post your article in a similar public access site, but we ask that you specify that the release date be no earlier than the date of publication on the *Spectrum* website.

Thank you for submitting your paper to *Spectrum*.

Sincerely,
Victor Gonzalez
Editor
Microbiology Spectrum

Reviewer #1 (Comments for the Author):

Reviewer's comments

Synergistic Effect of Combination Mucoactive Drugs with Phages against *Pseudomonas aeruginosa* and *Klebsiella pneumoniae* by Bingrui et al. investigates the potential synergy between N-acetylcysteine (NAC) or ambroxol hydrochloride (AMB) and phage therapy against *Pseudomonas aeruginosa* and *Klebsiella pneumoniae*.

Their results indicated that the highest concentrations of NAC (2 mg mL⁻¹) and AMB (2.5 mg mL⁻¹) demonstrated inhibitory effects on the growth of the bacteria, but lower concentrations of NAC and AMB showed little to no impact on bacterial growth. Their combination of phage and NAC indicated that of synergy in antibacterial effect. E.g. combining NAC with phage phiPA10 against strain ZS-PA-35 was significant. The authors also demonstrated that NAC affects biofilm formation in *P. aeruginosa* and *K. pneumoniae*.

The authors utilized a scientifically sound approach for this study, which included Bacterial susceptibility to NAC/AMB in LB medium, Phage-NAC/AMB combination assay, The impact of mucoactive agents on phage stability, Antiviral activity of

NAC/AMB on OC43 assay, Phage adsorption assay, Real-time quantitative PCR, Biofilm formation and twitching motility assay. With the profound implication of these findings to medicine and phage biology, I recommend the publication of this manuscript at its current state.

Reviewer #2 (Comments for the Author):

I appreciate the chance to review Synergistic Effect of Combination Mucoactive Drugs with Phages against *Pseudomonas aeruginosa* and *Klebsiella pneumoniae* by Sui et al.

Overall the manuscript is well written, and the data is presented well. There are a few minor points of clarification such as in the abstract line 20 the use of "two isolates of MDR Pa and Kp" led me wondering if there were 4 isolates total or 2 isolates, one of each, so perhaps clarify that point.

Care must be taken to insure that the OD plots are readable, in current form the text is very small.

Reviewer's comments

Synergistic Effect of Combination Mucoactive Drugs with Phages against *Pseudomonas aeruginosa* and *Klebsiella pneumoniae* by Bingrui et al. investigates the potential synergy between N-acetylcysteine (NAC) or ambroxol hydrochloride (AMB) and phage therapy against *Pseudomonas aeruginosa* and *Klebsiella pneumoniae*.

Their results indicated that the highest concentrations of NAC (2 mg mL⁻¹) and AMB (2.5 mg mL⁻¹) demonstrated inhibitory effects on the growth of the bacteria, but lower concentrations of NAC and AMB showed little to no impact on bacterial growth. Their combination of phage and NAC indicated that of synergy in antibacterial effect. E.g. combining NAC with phage phipa10 against strain ZS-PA-35 was significant. The authors also demonstrated that NAC affects biofilm formation in *P. aeruginosa* and *K. pneumoniae*.

The authors utilized a scientifically sound approach for this study, which included Bacterial susceptibility to NAC/AMB in LB medium, Phage-NAC/AMB combination assay, The impact of mucoactive agents on phage stability, Antiviral activity of NAC/AMB on OC43 assay, Phage adsorption assay, Real-time quantitative PCR, Biofilm formation and twitching motility assay.

With the profound implication of these findings to medicine and phage biology, I recommend the publication of this manuscript at its current state.